# Peptide–Protein Interactions: From Drug Design to Supramolecular Biomaterials

**DOI:** 10.3390/molecules26051219

**Published:** 2021-02-25

**Authors:** Andrea Caporale, Simone Adorinni, Doriano Lamba, Michele Saviano

**Affiliations:** 1IC-CNR, c/o Area Science Park, S.S. 14 Km 163.5 Basovizza, 34149 Trieste, Italy; doriano.lamba@ic.cnr.it; 2Dipartimento di Scienze Chimiche e Farmaceutiche di Università di Trieste, Via L. Giorgieri 1, 34127 Trieste, Italy; SIMONE.ADORINNI@phd.units.it; 3Istituto Nazionale Biostrutture e Biosistemi, Consorzio Interuniversitario, Viale delle Medaglie d’Oro 305, I-00136 Roma, Italy; 4Istituto di Cristallografia, Consiglio Nazionale delle Ricerche (IC-CNR), Via Giovanni Amendola 122/O, 70126 Bari, Italy

**Keywords:** peptidomimetic, molecular recognition, peptide design, amyloids, peptide/protein interactions (PPI), self-assembling peptide (SAP), drug delivery, supramolecular biomaterials

## Abstract

The self-recognition and self-assembly of biomolecules are spontaneous processes that occur in Nature and allow the formation of ordered structures, at the nanoscale or even at the macroscale, under thermodynamic and kinetic equilibrium as a consequence of specific and local interactions. In particular, peptides and peptidomimetics play an elected role, as they may allow a rational approach to elucidate biological mechanisms to develop new drugs, biomaterials, catalysts, or semiconductors. The forces that rule self-recognition and self-assembly processes are weak interactions, such as hydrogen bonding, electrostatic attractions, and van der Waals forces, and they underlie the formation of the secondary structure (e.g., α-helix, β-sheet, polyproline II helix), which plays a key role in all biological processes. Here, we present recent and significant examples whereby design was successfully applied to attain the desired structural motifs toward function. These studies are important to understand the main interactions ruling the biological processes and the onset of many pathologies. The types of secondary structure adopted by peptides during self-assembly have a fundamental importance not only on the type of nano- or macro-structure formed but also on the properties of biomaterials, such as the types of interaction, encapsulation, non-covalent interaction, or covalent interaction, which are ultimately useful for applications in drug delivery.

## 1. Introduction

The molecular self-assembly process can be defined as a reversible and spontaneous organization of some molecular entities (small or macromolecules, peptides, or proteins) under thermodynamic equilibrium into a well-defined and stable arrangement without external interventions [1,2]. These processes occur in the assembly of a variety of functional sub-cellular structures in physiological as well as pathological processes. Examples range from hemoglobin and membrane channels for transport, to amyloids associated with Alzheimer’s, Parkinson’s, diabetes, or amyotrophic lateral sclerosis diseases [3,4,5].

In the field of peptide/protein and/or protein/protein interaction (PPI), the supramolecular association achieved through weak intramolecular forces is a key process in both chemical and biological recognition, whereby the interaction may be associated with a conformational change leading to the activation of a biological pathway [6,7]. The understanding of the supramolecular interaction between a ligand (e.g., a small peptide) and its target (e.g., the surface of a protein or receptor) is a strategic approach to understand the mechanisms triggering biological communication [7] and to design new drugs [8]. From a physico-chemical point of view, assembling processes are mediated by weak, non-covalent interactions at macroscopic, microscopic, and nanometer scales. They include hydrogen bonds, electrostatic, hydrophobic and van der Waals interactions and π-π stacking [9,10,11]. For example, the driving forces used to favor the formation of fiber-like structures are based on the hydrogen bonds [12] or π-π interactions [13]. If the dissociation energies are compared, a homolytic dissociation of covalent bonds requires 100–400 kJ/mol; meanwhile, that of hydrogen bonds only requires 10–65 kJ/mol. The assembling process needs often a combination of multiple interaction sites [14,15]. A great number of organic molecules have been reported to be able to form supramolecular polymers in a controlled way. The fibrillation process, such as amylogenesis in Alzheimer’s disease, is probably the most striking biological example of peptide self-assembling. From those studies, new technologies started, particularly for a variety of biomedical applications, most notably as scaffolds for regenerative medicine, drug delivery systems, tissue engineering, and defined matrices for 2D and 3D cell culture.

A special class of self-assembling and self-recognition molecules are peptides [16] and, recently, peptidomimetics [17]. The research on self-assembling peptides has started since the early 1990s. The range of applications of self-assembling peptides covers new biological materials, surface coatings, and semi-conducting devices, as well as a new class of antimicrobial agents to prevent and combat antibiotic resistance [18]. Self-assembling peptide (SAP) systems involve natural or synthetic scaffolds that are capable of presenting multiple cell-interactive components in spatially resolved networks via supramolecular self-assembly or spontaneously ordered aggregates [19]. Hydrogen bonding, hydrophobic interactions, electrostatic interactions, and van der Waals forces allow maintaining the peptide-based self-assembled structures in a stable low-energy state [20]. Thus, the self-assembling processes are influenced by several factors: (i) amino acid sequence, (ii) the degree of hydrophobicity, (iii) the length of the peptides, and (iv) the self-assembling time [21]. These new materials based on SAPs involve synthetic scaffolds capable of containing multiple cell-interactive components and often yielding hydrogels [22] able to mimic the extracellular matrix [23] for applications in a variety of biomedical fields [24]; some examples as reported in Table 1 [25,26,27,28,29,30,31,32,33,34,35,36,37,38,39,40,41,42,43,44,45,46,47,48,49,50,51,52,53,54,55,56,57,58,59,60,61].

Moreover, peptides provide several advantages in the form of multifunctionality, multivalency, synthetic definition, molecular specificity, and control over the nanoscale positioning of ligands and other biomolecular features. However, prediction of the supramolecular behavior of SAPs is a challenge because a large number of factors are involved in the self-assembling process [62]. Nevertheless, fine control over the process allows for innovative solutions in drug delivery [63], which included the delivery of proteins [64,65] and peptide-based therapeutics [66,67], tissue engineering [68,69], biomimicry [70], cancer cell detection [71], and even vaccine adjuvants to stimulate the immune response [72], through their use as emulsifiers [73], pigments [74], catalysts [75], and semiconductors [76]. Several studies have demonstrated the crucial role of the SAP secondary structure to control the supramolecular architecture [66]. Although this topic has already gathered the interest of many researchers, as highlighted in recent reviews [8,22,23,24,62,66,68,77], here, we report on some recent developments in the application of short peptides both as molecular probes to map the peptide-protein and ligand-receptor interacting surface of biological relevance (molecular recognition process) as well as their applications in biomedical materials (hydrogels, specially).

## 2. Peptides as “Tools” for (Bio)supramolecular Interaction

The non-covalent interactions that undergo supramolecular processes, such as hydrogen bonding, electrostatic attraction, and van der Waals forces, underlie the formation of secondary structures such as α-helix and β-sheet, which play a fundamental role during biological molecular recognition and/or self-assembly. For example, the β-sheet structure is well known for its occurrence in amyloid fiber involved in several neurodegenerative diseases [78]. On other hand, functional amyloid fibrils are often found in the extracellular matrix or on the cell surface of various microorganisms, where they generally serve protective roles [71,79,80]. Thus, a large number of investigated artificial peptides were designed to fold into the β-sheet secondary structure, and their self-assembled nanostructures have been applied for regenerative tissue engineering and the design of functional biomedical materials [10,81,82]. For these reasons, exploiting the secondary structure conformation is of particular relevance in order to understand the hierarchical process of self-assembly [71,83]. Four different classes [84,85] have been proposed: (1) α-helix [86], (2) peptide-amphiphile (PA) [87], (3) β-sheet [88], and (4) collagen-based systems (Figure 1) [89].

Moreover, small peptides can be also used as “tools” in the drug design, optimization, and discovery of new biologically active molecules in accordance with hierarchical strategies [93,94,95,96,97]. Here, we report some selected examples taken from the literature for each class to describe the rational approach of hierarchical strategies in the field of supramolecular science and self-assembling or molecular recognition.

### 2.1. α-Helix

In α-helices, the stabilization of the peptide backbone is based on intramolecular hydrogen bonds between the oxygen of the carbonyl group and the hydrogen of every third amide group [98]. This arrangement extends outward from the side chains of amino acids from the surface of each helix turn. Although a single α-helix is in itself not a thermodynamically stable conformation, a more stable structure can be achieved by assembling together α-helices in coiled coils [99]. For instance, integral membrane proteins, G protein-coupled receptors (GPCRs), form a large group of evolutionarily related proteins that are cell surface receptors, which detect molecules outside the cell and activate cellular responses. GPCRs mediate a wide variety of physiological processes, including the self-recognition, through supramolecular binding, between receptor and peptide/ligand. In terms of structure, GPCRs are characterized by an extracellular N-terminus, followed by seven transmembrane α-helices connected by three intracellular and three extracellular loops and finally an intracellular C-terminus. These receptors undergo conformational changes that facilitate the binding and activation of multiple effectors [100]. In this field, Caporale et al. reported a hierarchical approach based on the hypothesis that a stable, amphipathic α-helix of the N-terminal fragment of the parathyroid hormone was a prerequisite for the binding and activation of its receptor (PTH1R). The role of side chains of the residues has been investigated [101], and a molecular model of interaction has been proposed to account for the gained information and the precise studies on pharmacophore amino acids in position 8 [102] and 2 [103] of the mutated N-terminal fragment PTH(1—11) [90,104]. In the molecular model, the molecular recognition by the N-terminal segment of PTH appeared not merely because of helical stabilization, but it also depended on the interaction of charged/hydrophobic/hydrophylic surfaces with the extracellular domain of the receptor [103,104,105,106]. In the field of self-assembly, a typical case is the coiled-coil structure [107] of α-helices. The coiled coils are self-assembling structures often found in numerous proteins in the cytoskeleton and the extracellular matrix [108]. They are characterized by two (or more) α-helices, with a repetitive pattern of seven amino acids, which is called heptad (*abcdefg*)_n_. Peptides conforming to these rules will form into a right-handed α-helix, assembling into helical bundles with left-handed supercoils [109]. Positions *a* and *d* form the hydrophobic core of the super-helical structure; *e* and *g* are mostly charged amino acids, which participate in interhelical salt bridges; *b*, *c,* and *f* are solvent-exposed, often polar amino acids, which contribute to the helix propensity of the individual helices [110,111,112]. Consequently, the combination of helix propensity and hydrophobic core packing determines the mechanical stability of coiled coils. From an energetic point of view, there is a balance between contributions of helix propensity and hydrophobic core packing, and both should be considered when the mechanical properties of coiled coils are evaluated for applications [91].

Mondal et al. [113] proposed a short synthetic peptide containing C α-tetra-substituted amino acids [114], H_2_N-Phe-Aib-Leu-Glu-Aib-Leu-Phe-OH, as capable of self-assembling into a coiled-coil structure in which the helical interface was stabilized by supramolecular knob-into-hole packing [71], as shown by single-crystal X-ray crystallography, fiber diffraction, and NMR spectroscopy. The peptide self-assembled into rod-like super helical nanofibers that formed lyotropic liquid crystals with predictable thermotropic activities. They envisaged an application in areas related to liquid crystalline materials. Biotechnological applications of coiled-coil proteins and their design principles based on the modularity of fragments, which mimic Nature’s complex self-assembling systems spanning from the nanoscale to the macroscale [115], were described recently by Lapenta et al. [116,117].

Over the last decade, Koksch’s research group [85,118,119,120] designed a class of peptides based on α-helical coiled-coil containing structural elements required for both stable α-helical folding and β-sheet formation. They are thought of as model peptides that are able to undergo structural transitions from an α-helical structure to β-sheet rich amyloid fibrils when they are exposed to changes in the physico-chemical environment, such as pH, ionic strength, or the presence of metal ions, resulting in conformational changes and distinct aggregate morphology [121]. For example, under acidic conditions, they observed the formation of either typical amyloid-like fibrils or extended α-helical fibers [122]. Recently, they used a pH-switchable model peptide to rationalize the lack of extension of fibrillation in the third dimension. They observed unfavorable intermolecular electrostatic repulsions between charged residues of the peptides along their solvent-exposed surfaces. Thus, a fibrillar structure propagating in only two dimensions arose, which was in agreement with other recent structural studies of fibrillar systems that are able to form a highly regular carpet-like superstructure [123]. Recently, during the study of the folding process and stability of PREP1 (PBX-regulating protein-1) full-length protein, which is involved in growth and differentiation during vertebrate embryogenesis [124], Doti et al. [125] identified two regions, PREP1(297–311) and PREP1(117–132), that were able to form amyloid-like oligomers in conditions close to physiological. In their study, they observed, for the first time in a native protein, that the PREP1(117–132) fragment was able to adopt either α-helical or amyloid-like β-rich states depending on the environmental conditions (pH and temperature) by adopting a chameleon-like behavior, as previously observed by Koksch et al. [122] in their model peptides. On the contrary, the PREP1(297–311) fragment exhibited a strong tendency to form amyloid-like β-rich oligomers in all conditions of pH and temperature. They interrelated that observation with the thermostability of full-length PREP1. Moreover, they observed through a hierarchical strategy that no residue in the PREP1(117–132) fragment was essential for the formation of amyloid-like assemblies, because it was only when all leucine/isoleucine residues were Ala-substituted that the β-amyloid-like assemblies were destabilized. These observations were in line with some recent studies by Bera et al. [126,127]. They classified the amino acids side chains β-sheet propensity based on various criteria, such as hydrophobicity, steric bulkiness, and folding. They calculated the contact number, which is defined as the total number of contacts divided by the total number of molecules, in order to analyze the molecule-molecule interactions for the aggregation propensity of Phe toward Ile and Gly. They found that the order of interaction strength from strongest to weakest is Phe-Phe > Ile-Ile > Gly-Gly, confirming the importance of hydrophobic interactions [127].

### 2.2. β-Sheet

Another type of secondary protein structure is the β-strand, which is characterized by a large number of intermolecular hydrogen bonds between the peptides -C=O—H-N-. Each β-strand is connected laterally by hydrogen bonds, forming a pleated sheet [128], which is able to rigidify the structure precisely through the interpeptide and interchain hydrogen bonds [86,87]. The hydrogen binding patterns can result in two different forms: parallel or antiparallel. The antiparallel β-sheets are energetically more favored than the parallel forms because the hydrogen bonds are better aligned [129,130]. These structures form both indefinite and definite assemblies. Indefinite assemblies are peptide fibers with hundreds of nanometers to a few micrometers in length [131,132]. The introduction of d-amino acids can be used as a means to stabilize turn conformations, which can self-assemble into tubular water channels as wide as a few nanometers [71]. On the other hand, vesicles and micelles with discrete dimensions are examples of definite supramolecular assemblies [66]. An example of antiparallel self-assembly is the β-sheet structure, which self-assembles very commonly, facially or laterally, into hydrogels, with applications in different fields [55,133,134,135]. Hydrogels generated by ^D^Phe-^L^Phe-^L^Leu-^L^Asp-^L^Val containing the Leu-Asp-Val sequence to activate β1 integrin have been shown to mimic successfully fibronectin of the extracellular matrix, displaying high cell viability, adhesion, and spreading [55]. Recently, Garcia et al. have reported high cell viability for fibroblast and keratinocyte cells with a mild antimicrobial activity against *E. coli* when they used *N*-(4-nitrobenzoyl)-Phe as a self-assembling molecule [135]. Moreover, they reported that the hydrogel is thermos-reversible and disassembles within a range of temperatures features under fine-tuned experimental conditions. It is interesting to show how the amino acid stereo-configuration of the diphenylalanine motif is able to generate dramatic changes in the nanostructures in the hydrogels features under physiological conditions, with the formation of hydrogels that structure from amorphous aggregates to ordered and stable hydrogels [134].

Amyloidogenic peptides are able to form β-sheets and self-assemble and are an object of an increasing interest for medical and pharmaceutical aspects, in particular “β-sheet breakers” (BSB) that are able to inhibit Aβ aggregation [136,137,138,139]. Amyloid self-assembly plays a crucial role in the field of diseases caused by protein misfolding and aggregation into fibrils (Alzheimer’s disease (AD), Parkinson’s disease (PD), Huntington’s disease (HD), type 2 diabetes (T2D), prion protein (PrP)-related encephalopathies, and many other amyloidosis [140,141]. These amyloid aggregates are characterized by the β-sheet structure, and they are organized into long fibers with the protein backbone orthogonal to the fiber axis [142].

In the last decade, an approach based on hierarchical strategies of peptide drug design was proposed to inhibit amyloidogenic self-assembly of Islet Amyloid PolypPeptide (IAPP) (Figure 2). IAPP is a protein secreted by the islet beta cells that are stored with insulin in the secretory granules and released in concert with insulin. Normally, IAPP modulates insulin activity in skeletal muscle, influencing energy homeostasis, satiety, blood glucose levels, adiposity, and even body weight. IAPP amyloid is found in insulinomas and in the pancreas of many patients with diabetes mellitus type 2. IAPP could be toxic at the competitive or non-competitive level. IAPP analogs are now being investigated in the treatment of DM2 and obesity [143,144].

In particular, molecular strategies to interfere with amyloid formation using the high affinity IAPP–Aβ cross-interaction to block the generation of the amyloidogenic protein through inhibition of primary and/or secondary nucleation has been exploited [145]. The islet amyloidosis and pancreatic β-cell apoptosis are two key determinants of islet dysfunction in type 2 diabetes supported by in vivo studies [146,147,148]. An amyloid core(22–27) segment—NFGAIL—of the IAPP sequence with a double N-methylation, IAPP-GI, has been reported to be non-amyloidogenic and effectively suppressed IAPP amyloid self-assembly and cytotoxicity [149,150] and inhibited Aβ40 cytotoxic self-assembly, blocking the inflammatory processes mediated by Aβ and IAPP aggregates in microglial cells and macrophages [151].

A recent study on hydrophobic and π-π interactions in NFGAIL self-assembly based on fluorinated and iodinated phenylalanine analogues revealed a synergy between the aggregation behavior and hydrophobicity of the phenylalanine residue. These observations suggested an amyloidogenic behavior of NFGAIL independent from π-stacking geometries [157]. A systematic study on the molecular basis of Aβ-IAPP interaction resulted in the identification of short Aβ and IAPP peptide sequences as hot regions of the Aβ-IAPP hetero-cross-interaction interface. The resulting peptides showing affinities in the nano- to low-micromolar range based on membrane-bound peptide arrays of 10-residue Aβ(1–40) sequences covering full-length Aβ40 are positional shifted by one residue [152]. The role of IAPP individual regions, i.e., the N-terminus (1–7), the C-terminus (30–37), and IAPP(8–28), which contains the two hot spots, IAPP(8–18) and IAPP(22–28), and their inhibitory effect on Aβ40 self-association into cytotoxic aggregates and fibrils has also been addressed [153]. The N-terminus of IAPP is necessary for the potent inhibitory effect during the interactions between the IAPP hot regions (IAPP(8–18) and IAPP(22–28)) and Aβ-IAPP. Moreover, it is likely that the N-terminus of the IAPP fragment is able to stabilize a specific conformation in IAPP that is able to interact with Aβ(1–40). In order to understand the mechanism to block the amyloid self-assembly of Aβ, IAPP, or both polypeptides, the role featured by either “hot segments”, IAPP(8–18) and IAPP(22–28), in inhibition caused by IAPP-GI during self- and its cross-interaction Aβ/IAPP [152] and hot-segment-linking sequence in stabilizing the conformation of IAPP-GI, has been investigated [154]. The hot-segment-linking approach leads to the design of a library of IAPP-GI cross-amyloid interaction surface mimics (ISMs). They exhibited different amounts of β-sheet/β-turn structures, which showed an inhibitory effect because of the high-affinity binding toward prefibrillar Aβ(1–40) or IAPP species, resulting in their sequestration from amyloidogenesis in the form of amorphous and non-toxic hetero-assemblies [158,159]. The contribution of the single aromatic/hydrophobic residues, Phe15, Leu16, Phe23, and Ile26, within the amyloid core IAPP region as hot spots or key residues of cross-interaction with both IAPP self- and its hetero-assembly with Aβ(1–40) has been established by using a systematic Ala-scan approach, fluorescence spectroscopy, and other biophysical methods. In particular, the Phe15 and/or 23 are required for IAPP interaction with Aβ(1–40) but not with self-interaction with IAPP. Four key residues have been hypothesized to play a crucial role in the stability of prefibrillar Aβ-IAPP hetero-assemblies likely in the context of previously proposed β-strand-loop-β-strand IAPP conformers [155]. Recently, ISMs [160] through solution-state and solid-state NMR in combination with ensemble averaged dynamics simulations and other biophysical methods have been characterized. The best candidate, a peptide containing a charged hot-segment-linking sequence with a double N-methylation in positions 23 and 26, R3-GI, adopted a β-like structure where the peptide conformer was cis, suggesting a turn structure induced by proline [161]. ISM assemblies provided a multivalent surface for interactions with Aβ(1–40), resulting in its sequestration into off-pathway non-toxic aggregates similar to colloid assemblies. All collected data [156] converged in the design of macrocyclic peptides capable of mimicking IAPP surfaces interaction maintaining only minimal IAPP-derived self-/cross-recognition elements identified before [154,155]. The approach pursued was based on conformational restriction by cyclization, sequence truncation, and multiple amino acid substitutions with Gly or D-configured residues, in order to increase proteases resistance. The β-hairpins/β-sheet folds of IAPP were kept due to the likely prerequisite for its inhibitory function during the supramolecular surface interaction. This approach led a nanomolar Aβ(1–40)-selective inhibitor presenting high proteolytic stability in human plasma and human Blood-Brain Barrier crossing ability in a cell model.

Another “β-sheet breakers” (BSB) example based on a modified fragment of Aβ(16–20), Leu-Pro-Phe-Phe-Asp [162,163] conjugated with trehalose, was able to slow down the Aβ aggregation process, has been reported [164]. In order to engineer amyloidal peptide self-assembly and obtain new amyloidal nanostructures, the halogenation could be used. Nanostructures based on the Aβ(16–20) peptide-derived core-sequence Lys-Leu-Val-Phe-Phe by controlling the number, position, and nature of the halogen atoms introduced into either one or both phenylalanine benzene rings have been described [165,166]. Molecular dynamics simulations on different halogenated derivatives of the Aβ(16–20), by using a modified AMBER force field to include an σ-hole located on the halogen atom with a positively charged extra particle, well agree with the crystallographic data [165] and experimental results [166]. In particular, the analysis showed that the formation of halogen bonds [167] stabilizes the supramolecular structures, allowing further possibilities for a fine control of the amyloid nanostructure [168]. Halogenation proved useful also to achieve fine control over the hierarchical assembly of heterochiral Phe-Phe to yield homogenous fibrils and hydrogels, although no halogen bonding was involved in this case [56]. Interestingly, heterochirality per se allowed substituting inter- with intra-molecular interactions, thus stabilizing 4 nm-wide nanotubes composed of two peptide layers around an inner water channel, and overall alleviating the homochiral Phe-Phe microtubes toxicity observed in cell culture [56].

### 2.3. Peptide Amphiphile (PA)

Amphiphilic peptides (PAs) have interesting properties, such as surfactant-mimicking properties and bioactive functions that are able to generate ordered, self-assembled nanostructures for applications in drug delivery, tissue engineering, or smart hydrogels modulated by changes of pH, light, ionic strength, and temperature [71,169,170,171,172,173,174]. For biocompatibility, specific biological functions, and low toxicity toward normal cells and tissues, PAs have been recently considered as ideal drugs carriers. The nanocarriers derived from PAs are able to control drugs release as well as enhance cell uptake in response to either the stimulus of the physiological environment or to specific biological factors in the location of the lesion [175]. PAs that self-assemble into hydrogels have been reported to mimic the extracellular matrix, which is in particular applied to the mimicry of the nanoscale structure of bone [176]. They built PA using a peptide domain adjacent to the lipid tail with a high propensity to form β-sheets, which is followed by charged residues to ensure solubility in water, including spacers in order to allow flexibility when PA is linked to a bioactive molecule. In aqueous environments, the hydrophobic lipid tails lead to lipid aggregates, while hydrogen bonding between peptide segments yield “filamentous” assemblies with β-sheet domains. The observed structures are cylinders, ribbons, twisted, or fibrous aggregates [177]. The nanostructures find application in various fields from regenerative medicine to drug delivery [170,178]. Nanofibers derivatized with the sequence Ile-Lys-Val-Ala-Val, neurite-promoting laminin epitope are able to induce a quick differentiation of cells into neurons while discouraging the development of astrocytes [179]. Similarly, the nanofibers can be applied as cancer therapeutics. For example, Standley et al. [178] synthetized PAs containing a fragment of (Lys-Leu-Ala-Lys-Leu-Ala-Lys)_2_, which is able to induce the cancer cell death by membrane disruption. PA self-assembles into bioactive, cylindrical nanofibers that induced breast cancer cell death by a mechanism associated with membrane disruption. Webbers et al. [180] reported the synthesis of nanofibers based on peptide amphiphiles self-assembling in nanoscale filaments decorated on their surface with a Vascular Endothelial Growth Factor (VEGF)-mimetic peptide in order to develop a novel therapy for ischemic cardiovascular disease. They observed proangiogenic behavior in endothelial cells, which was indicated by an enhancement in the proliferation, survival, and migration in vitro. Recently, an accurate review by Stupp’s group was published [181]. Another interesting example of PA was presented, which studied the self-assembly behavior of the short peptide A1H1, H-Ala-Ala-Thr-Ala-Val-Ser-His-Thr-Thr-His-His-Ala-OH, which is one of the most abundant β-sheet forming domains within the suckerin protein family [182]. The observed formation of chiral hollow core-shell cylinders was due to the presence of a polar and bulky domain (-His-Thr-Thr-His-His-Ala-) and of another compact and apolar domain (-Ala-Ala-Thr-Ala-Val-Ser-). Thus, the polar region was exposed to the solvent molecules, assembling into fibrillar supramolecular chiral aggregates with helical ribbon configuration in solution. When the concentration increased, the fibrils gelled while preserving the same mesoscopic features. These studies offered a way to expand the understanding of self-assembling peptides with β-sheet forming propensity under different conditions. This peptide might be attractive as a nanoscale drug carrier with pH-dependent release [182].

### 2.4. Poly(L-Proline) Type II (PPII) Helices

An interesting structure, typical for proline-rich peptides, is poly(l-proline) type II (PPII) helix, which is a left-handed helix with three amino acids per turn [91]. The PPII helix is assembled into a triple helix in collagen and is able to contribute to the structural stability [183]. In particular, the PPII helix backbone is exposed and available for intermolecular hydrogen bonds, leading to the stabilization of self-assembled oligopeptide vesicles [184]. The PPII helix has a characteristic rigidity and well-defined secondary structure derived from the innate features of proline, which is the only naturally cyclic amino acid, generating often-populated trans and cis conformers around the tertiary amide bonds formed in proline oligomers. For these reasons, oligo-prolines are applied as a “molecular ruler” to define a distance and/or as a “molecular scaffold” with precisely located and predictably oriented substitutions along the polymeric backbone [185]. Oligo-prolines do not self-assemble on their own, but they serve as a scaffold for distance control over the number and spatial preorganization of π systems [109]. Recently, interesting applications were proposed by expanding the utilization of oligoproline-chromophore conjugates into advanced topologies. Their insights were based on building blocks oligoproline-perylene monoimides (PMI) conjugates. The oligoproline scaffolds were able to guide the hierarchical self-assembly of sterically demanding electron-poor (PMI) chromophores into a variety of unprecedented nanostructures (Figure 3). These building blocks formed fibers via π-π stacking of the N- and C-terminal chromophores in which full and empty spaces alternated at regular intervals in a defined structure [186,187,188]. In this macromolecular structure, the polyproline II plays the role of “molecular scaffold” functionalized with PMI capable of assuming a left-handed structure that is conformationally well defined. Macroscopically, the oligoproline-PMI conjugates organize themselves into an intercalated double layer in which PMI are in the grooves of the oligoproline scaffold, forming a “sandwich” structure, with an inner hydrophobic part between two hydrophilic polyproline scaffolds. A cooperative process in solution has been reported very similar to macroscopic woven materials, giving mechanical strength and stability at the nanoscale. The role of the oligoproline scaffold was to form this self-assembled organic weave through precise spatial control at the molecular level. The presence of four azide-prolines in the sequence of polyproline allows modulating the functionalization of hydrophobic self-assembling moiety of macromolecules. Recently, the conjugation of a quaterthiophene moiety via a triazole linkage with oligoprolines allowed having biocompatible molecules with a variety of well-ordered, open chiral, self-assembled nanostructures [189]. The applications of oligothiophenes in the development of organic materials are important in the field of organic electronic devices [190]. Interestingly, supramolecular assemblies, including nanostructured sheets, curls, or ribbons with variable length of the oligoproline moiety, could shape the assembly of the quaterthiophenes into parallel or antiparallel π-π stacked aggregates and could affect the molecular organization of the conjugates from monolayers to double- layered sheets and helically twisted ribbons. As the performance of organic electronic devices depends on their molecular organization in the solid state [191,192], the study provided guidelines for the rational design of related conjugates with predictable self-assembly properties and a combination of electron-donor and electron-acceptor components for supramolecular assemblies with unique molecular arrangements [186,187,188,193].

## 3. Self-Assembling Peptides in Drug Delivery Applications

Short peptides and their derivatives constitute highly advantageous building blocks for the self-assembly of nanostructured biomaterials [194,195]. They present different advantages including ease and low cost of preparation, simpler characterization, great stability, and biocompatibility [196,197]. Recently, the interest in short peptides that are able to form hydrogels has advanced for the “bottom-up” approach to creating novel self-assembled biomaterials, as demonstrated by a recent computational study on a series of tripeptides generated by the known self-associating tripeptide, Ac-Ile-Val-Asp, which is used as a structural template. The features of studied tripeptides ranged from self-assembling into fibrils to hydrogelation [198,199]. Bera et al. showed that the tripeptide Pro-Phe-Phe over two weeks assembles into a helical-like sheet and that it is stabilized by the hydrophobic interfaces of phenyl side chains, as a core structural motif in amyloid fibrils. They observed also that the replacement of Pro with hydroxyproline generates minimal helical-like assemblies with mechanical rigidity [200]. Nowadays, hydrogel materials are widely used in several different areas, and drug delivery is probably one of keys used for such systems in the biological and medical field, which has been amply covered in a variety of recent reviews [201,202,203,204,205,206,207]. With a view to find new innovative nanotechnological solutions for the efficient and safe delivery of drugs, rapid progress is being made worldwide in this area of research [199,208,209]. For these reasons, we focus on medical applications of hydrogel generated by short peptides. The incorporation of drug molecules inside the nanostructures can be achieved by several means (Figure 4): (1) physical entrapment in the hydrogel; (2) non-covalent drug binding; and (3) covalent linkage of the drug. Here, some recent examples from the literature are reported.

### 3.1. Physical Entrapment

The most common and simplest method to formulate a bioactive compound using hydrogel as a carrier is through the physical entrapment of drugs in the gel’s matrix without chemical interactions [210]. The amphipathic hexapeptide H-Phe-Glu-Phe-Gln-Phe-Lys-OH was discovered to be a very promising hydrogelator in terms of the in vitro controlled release of model cargoes [58]. Due to the very low pH (< 4) of this first hydrogel, it is not suited for in vivo applications. In order to improve the self-assembly process and the in vivo stability, several changes on the chain of peptides have been carried out. The first modification regarded the introduction of acetyl and carboxamide groups on N- and the C-terminus of the peptide, respectively, in order to increase the gel’s pH compared to that of the original one. The amide at the C-terminus increased the gel pH, while the introduction of an acetyl group caused the precipitation of the compound, underlining the importance of a free amine for the self-assembly process. Another important modification that regards the stability in vivo of the hydrogel was the introduction of D-amino acids. From the library of the new synthesized peptide, two sequences, H-Phe-Glu-Phe-Gln-Phe-Lys-NH_2_ and H-Phe-Glu-Phe-Gln-Phe-Lys-NH_2_, presented the most promising physico-chemical and gelation properties, and in vitro and in vivo studies focused on these two sequences. The in vitro studies confirmed the resistance to the proteolytic degradation of the sequence with D-amino acid, for which no half-life could be determined. The in vitro studies’ release experiment, using the opiate drug morphine, revealed a burst effect followed by a sustained release over 3 days, resulting in a recovery of about 90%. This initial burst effect was closely linked to the nanomorphology of the hydrogel. The hydrogel consisted of large pores, and the drug was weakly incorporated to the gel network, causing fast initial release. This behavior could be very useful in those situations where it is necessary to provide immediate pain relief, followed by a prolonged therapeutic effect. Comparison between the two sequences, which differed only for the chirality of the amino acids, revealed an identical release; however, as regards the cytotoxicity, the all-D-sequence was not biocompatible, and therefore, it was not considered for any in vivo tests.

The synthesis of a tripeptide, Boc-Phe-Phe-Phe, with permuting *L*- and *D*-configuration has been reported [60]. Doxorubicin is a potent anticancer drug chosen for this study. The construction of a library of all possible stereoisomers allows a detailed study of the chirality effects on the self-assembly process, gel mechanical properties, and drug release. There are several side effects when it is administered in high dose, and so, the research on hydrogels as vehicle is a very promising research field. All the compounds, apart from the *LLD* and its enantiomer *DDL*, were found to form gels at physiological conditions, pH 7.46. The four stereoisomers *LLL, DLL, DDD*, and *LDD* successfully encapsulated the doxorubicin. The stability of these doped gels was examined in the presence of a proteolytic enzyme, and it was observed that gels made of the respective enantiomers *DDD* and *LDD* remained stable over time. The gel made of *LDD* was found to be the most efficient drug delivery vehicle with a maximum release of 76% after 79 h. Furthermore, the gel obtained from peptide alone did not show significant toxicity toward cancerous cells, while the corresponding doped gel killed breast cancer cells more efficiently than the free drug alone.

### 3.2. Non-Covalent Interaction

The entrapment of the drug through non-covalent interactions is an alternative method to achieve sustained drug delivery. The presence of a drug that participates in the building of superstructures influences their nanomorphology and their spectroscopic properties, paving the way of their use to reveal whether non-covalent interactions are effectively taking place. Moreover, the presence of interactions between the supramolecular structure and the drug could influence the release kinetics that could differ from those achieved through drug entrapment [199]. Marchesan and her group developed and studied a system where a poorly soluble drug was engaged in non-covalent interactions with an ultrashort peptide [61]. The peptide chosen was ^D^Leu-^L^Phe-^L^Phe, that is an established gelator in phosphate buffer at neutral pH, and recently, also anti-inflammatory drugs naproxen and ketoprofen were studied for their ability to engage in the supramolecular hydrogelation [210]. As regards the properties of the hydrogel, rheological experiments showed a lowered resistance to the applied stress of the hydrogel containing either drug, relative to the hydrogel of the tripeptide alone. This result suggested fewer connection points between the fibrils composing the soft material with the drug relative to the tripeptide alone. TEM (Transmission electron microscopy) analysis revealed thinner fibrils for the co-assembled system. Drug release studies highlighted differences between the two materials. Most of the naproxen was released within 72 h, whereas the release of ketoprofen was completed within 24 h, suggesting different interactions with the gel matrix, from which it was inferred that the naphthalene unit of naproxen was more efficient than the benzene ring of ketoprofen at engaging in non-covalent interactions with the aromatic units of the peptide. A follow-up study on the more hydrophilic 5-fluoruracil revealed the ability of the drug to interact only transiently with the peptide stacks (Figure 5), as its hydrophilicity resulted in notably faster release kinetics [211].

A promising vaccine adjuvant based on the supramolecular hydrogel physical carrier for a protein has been developed [212]. The peptides chosen were Nap-Gly-Phe-Phe-p-Tyr-OMe and its d-enantiomer that formed co-assembled nanofibers with ovalbumin, which was the model protein chosen for the study of immune response. The d-and l-peptide hydrogels induced a higher antibody production, relative to controls, and both of them had a good biocompability. Surprisingly, in the presence of ovalbumin, the d-gel showed the best performance, and it seemed to be a potential promising vaccine. The authors suggested that this enhanced efficacy could be ascribed to faster release kinetics due to weaker interactions between the antigenic protein and the fibers of the d-isomer.

### 3.3. Covalent Interaction

The covalent binding of drugs to a hydrogelator usually involves changing the properties of the self-assembling molecules (i.e., solubility, charge, presence of free carboxylic or amine groups at C- and N-terminus that could play a key role in the self-assembly process), and that makes this approach the most challenging. Generally, the drug is anchored to the peptide termini or to the amino acids’ side chains, and so, the release is achieved upon hydrolysis of the covalent linkage through biological or physicochemical triggers [199]. Chakroun et al. covalently linked the anticancer taxol to an amphiphilic short peptide to obtain an injectable hydrogel with precise and controlled drug release [213]. Four variations of the Paclitaxel (PTX) prodrug hydrogelator were designed, and all of them shared a common backbone: a linker, 4-(pyridin-2-yl-disulfanyl)butyrate, conjugating PTX to a peptide sequence consisting of an Arg-Gly-Asp-Arg sequence, a Val-Val segment to promote intermolecular hydrogel bonding among the conjugates, and a Gly-Gly spacer to improve the drug solubility. These self-assembling PTX prodrugs associate into filamentous nanostructures. Researchers proposed a two-phase release mechanism. The first phase involved network swelling and disruption due to a difference in osmolarity with the Phosphate Buffer Saline solution that induced the diffusion of the monomeric conjugates and the spreading of filaments caused by water that penetrated into the network. The second step occurred when the filaments dissociated into monomers releasing the drug. The release kinetics is dependent on the properties of the gels at the interface between the gel and the environment, which are slowed down by increasing the alkyl chain, or accelerated by adding oppositely charged amino acids. In 2016, Parquette developed a fascinating example where the antimetabolite 5-fluorouracil (5-Fu) was covalently linked to the shortest possible peptide, a dipeptide [214]. The dipeptide chosen was a dilysine that assembled into various nanostructures. Two different dipeptides were synthesized, and they differed regarding the type of linker between the peptide and 5-Fu: a self-immolative succinate polymer and a stable acetamide linkage. These two peptides self-assembled forming different nanostructures: nanotubes with a diameter of 16 nm and nanofibers with a diameter of 10 nm. The hydrogel formed by nanotubes exhibited a slow release of active 5-Fu. In vivo studies showed that the cytotoxicity of the nanotubes’ gel arose because of free 5-Fu’s release and the no-cytotoxicity of the peptide containing the stable linker. The choice of linkage also affected the strength, stability, and reversibility of the resulting hydrogels.

## 4. Conclusions and Perspectives

In the last 20 years, a remarkable expansion of useful nanostructure-based materials offered newer tools in the field of biological and biomedical sciences [215]. In particular, well-defined peptide self-assembled nanostructures, based on the non-covalent forces, have interesting advantages, for example good biocompatibility, low cost, tunable bioactivity, high drug-loading capacities, chemical diversity, specific targeting, and stimuli-responsive drug delivery at disease sites. Self-assembling peptides are able to give nanoparticles, nanotubes, nanofibers, and hydrogels. Their morphology and function can be changed at the molecular level by tuning the types and secondary structures of peptides, or by controlling external triggers such as temperature, pH value, and the electric field [215]. The rapid advance made in the biomedical application of nanomaterials, including bioimaging and drug delivery, shows their high potential. For example, some drugs loaded to peptide self-assembly nanomaterials by physical encapsulation or chemical conjugation methods enhance retention effects at tumor sites and consequently increase the uptake rate of drugs [216,217]. If peptides have a good biodegradability and biocompatibility, their features may change also when they become nanomaterials. In other words, self-assembled peptide nanostructures might have potential immune-stimulatory properties, which should be systematically investigated to avoid toxic effects on the renal system and the production of toxic amyloid fragments by nanomaterials. Thus, peptides and small molecule-based nanostructures might be intriguing alternatives for therapeutic delivery due to their good biocompatibility, easy design/synthesis and functionalization. In particular, it is still unclear as to why certain sequences gel, whilst others crystallize or precipitate under analogous conditions. Similarly, another area that needs further investigation is the understanding of the phenomenon of syneresis, whereby certain gels (but not others) over time release fluid. That process could have useful applications for the delivery of therapeutics, yet the structure-to-function relationship remains elusive to design ad hoc hydrogels that display syneresis. While the design and understanding of the self-assembly of bare self-assembling scaffolds has advanced in recent years, the decoration with bioactive ligands often leads to unexpected changes in the self-assembly. The unforeseen changes in the self-assembling structures upon ligand functionalization, including chirality, are a beautiful new entry to expand the scope of sizes, stabilities, and shapes of self-assembling systems [55,56,127]. These changes at the same time provide more molecular insights into these fascinating bioactive supramolecular architectures. Another challenge of peptide self-assembling nanostructures is the influence on the organisms, controlling the size and composition during processing, tenability, behavior in aqueous environment, stability, up-scaling, and degree of loading/entrapment of therapeutics. The future challenges to generate powerful new therapies in the regenerative medicine of biomaterials made from self-assembling, short peptides, and peptide derivatives include short-term goals such as integration with biopolymers and traditional implants, and long-term goals, such as immune system programming, subcellular targeting, bioactive signaling strategies, and the development of highly integrated scaffold systems and the development of more complex, hierarchical structures. Moreover, establishing the biocompatibility and immunogenicity of these nanostructures, as well as fine-tuning the physico-chemical properties by assimilating chemical modifications and optimizing the peptide functionality to minimize the toxicity without threatening their therapeutic activity are part of the future studies [218]. Finally, the knowledge and the control of microstructures can lead to the assembly of a complex macroscopic architecture with interesting functional properties of the resulting materials with potential for the healthcare applications. An important challenge in the supramolecular research, as recently demonstrated by Umerani [219], is to understand the direct correlation between the protein’s structural characteristics and macroscopic materials.

## Figures and Tables

**Figure 1 molecules-26-01219-f001:**
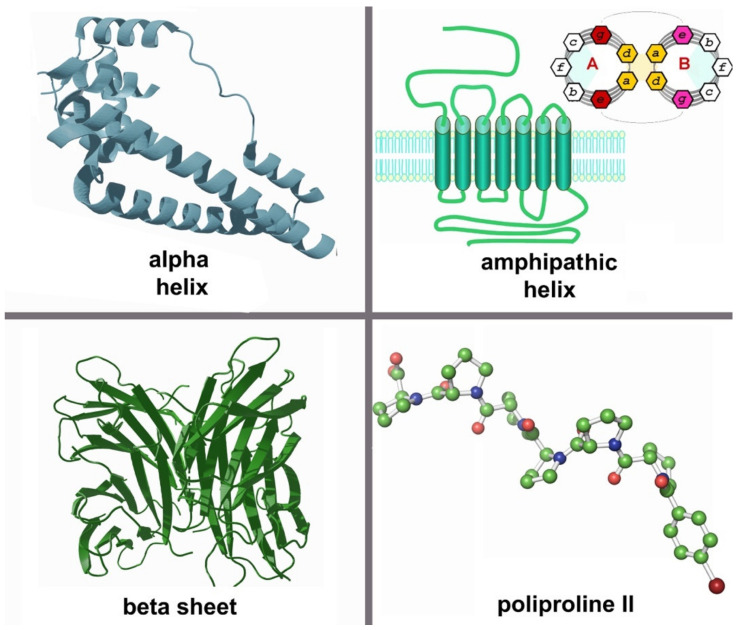
Secondary structures: α-helix (Protein data bank code: pdb1YG2); amphipathic helix (adapted by [90] and [91]); β-sheets (pdb5MAZ); polyproline II structure assumed by N-(4-Bromobenzoyl)hexaproline acetonitrile solvate [92].

**Figure 2 molecules-26-01219-f002:**
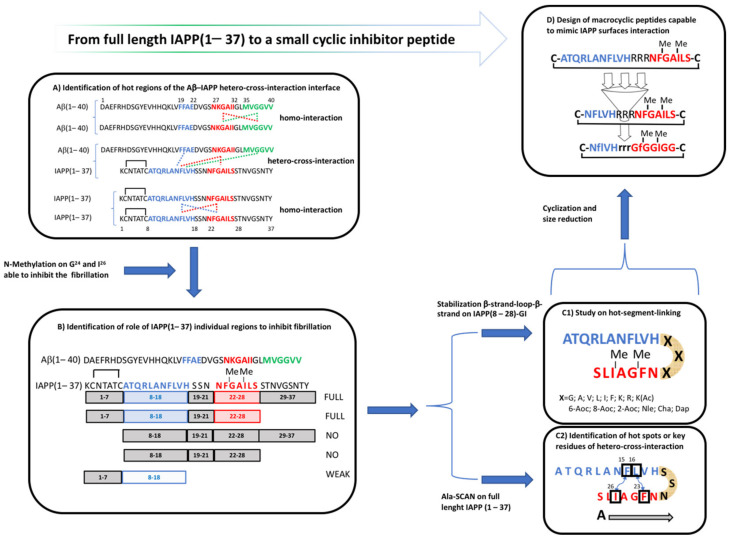
Kapurniotu’s hierarchical strategy applied on full length Islet Amyloid PolyPeptide (IAPP) (1–37) and short inhibitor fragment IAPP(8–28)-GI [149]: (**A**) Identification of hot regions of the Aβ-IAPP hetero-cross-interaction interface: summary of the determined cross- and self-interactions (solid arrows) between the hot regions (hot regions in blue, red, and green) of the Aβ-IAPP interaction interface [152]; (**B**) Role of IAPP individual regions: summary of the identified effects of the regions and shorter segments of IAPP-GI on Aβ40 amyloid formation. Regions in blue and red, or in light blue indicate regions or segments able to assemble both with Aβ40 and IAPP. Sequences in black indicate regions or segments that had no effect on the assembling processes [153]; (**C1**) Study on hot-segment-linking on IAPP(8–28)-GI [154]; (**C2**) Identification of hot spots or key residues of its hetero-cross-interaction and self-interactions between IAPP and Aβ [155]; (**D**) Design of macrocyclic peptides capable to mimic IAPP surfaces interaction [156].

**Figure 3 molecules-26-01219-f003:**
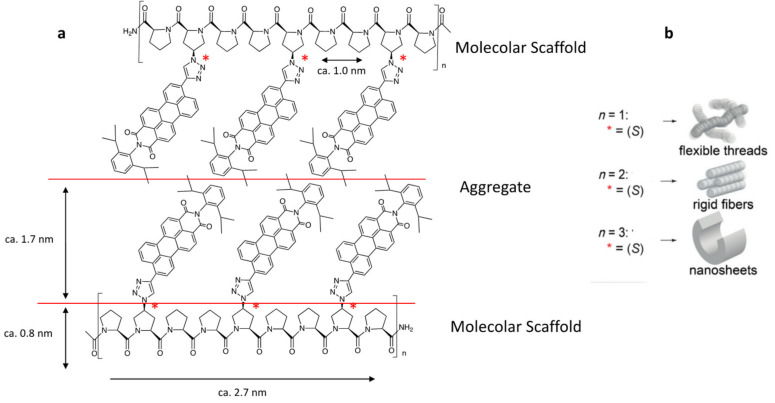
(**a**) General structure of oligoproline-perylene monoimide aggregate and the estimated molecular dimensions. It is interesting that a new stereocenter is introduced when a chiral perylene monoimide is bound with polyproline II (molecular scaffold) (**b**) models of supramolecular assemblies based on different units (adapted from [188]).

**Figure 4 molecules-26-01219-f004:**
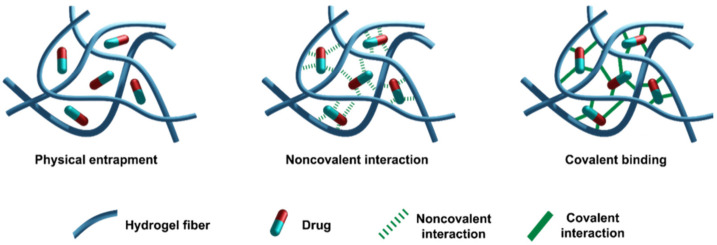
Different modes for drug entrapment.

**Figure 5 molecules-26-01219-f005:**
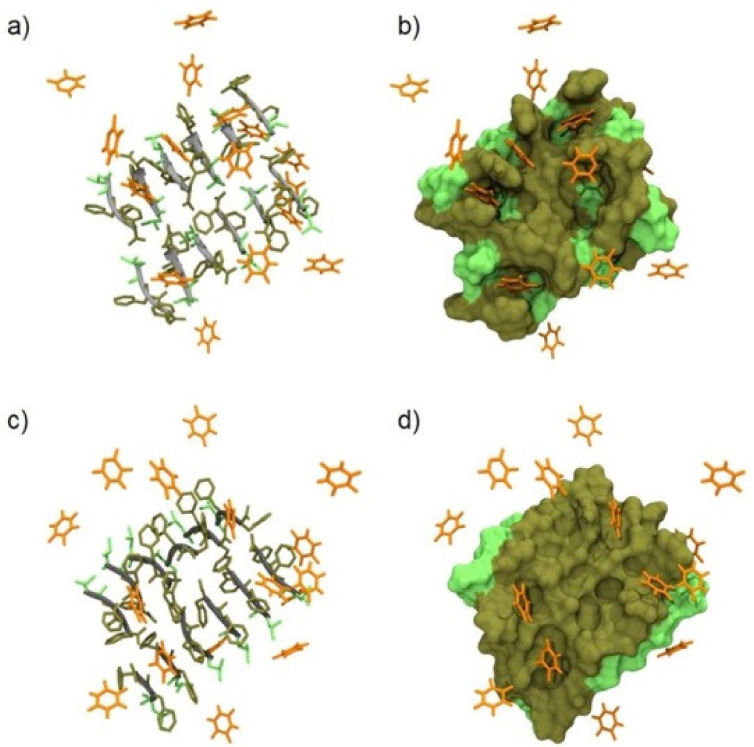
All-atom molecular dynamics snapshots (**left**) and molecular surface (**right**) of ^D^Leu-^L^Phe-^L^Phe stacks (gray/green) with 5-fluoruracil (orange). (**a**,**b**) and (**c**,**d**) images refer to antiparallel and parallel β-sheet conformations, respectively. Leu-side chains are depicted in light green, and Phe-side chains are depicted in dark green, respectively. Water and ions are omitted for clarity. Reproduced from ref. [211], https://doi.org/10.3390/gels5010005 (accessed on 17 February 2021), under the terms of the CC BY 4.0 license, https://creativecommons.org/licenses/by/4.0/ (accessed on 17 February 2021).

**Table 1 molecules-26-01219-t001:** List of some self-assembling peptide sequences and recent references (adapted from [18]). Here, the N-protected sequences, with Fmoc, Nap, Bzl, etc., were not reported for the sake of brevity.

Sequences	Applications	Ref.
VEVK9: VEVKVEVKV and VEVK12: VEVKVEVKVEVK combined with RGD	Increase fibroblast migration	[25]
Cyclo (CRGDKGPDC)	Tumor homing through encapsulation and release of photodynamic therapy drugs	[26]
Cyclo (RGDfK)	Drug targeting to RGD-α*_v_*β_3_ integrin interplay, for targeted cancer chemotherapy	[27]
Lyp-1: (CGNKRTRGC)	Drug targeting to gC1q receptor p32 protein for lymphatic metastases	[28,29,30]
C16V2A2E2K(Hyd)	Drug release of the drug nabumetone	[31]
V6K2: VVVVVVKK	Drug delivery of doxorubicin or paclitaxel to 4T1 mouse breast carcinoma cells	[32]
MAX8: VKVKVKVKVDPPTKVEVKVKV	Drug encapsulation and release of curcumin for brain tumor therapy	[33]
RAD/PRG/KLT RAD: Ac-(RADA)4-NH_2_ PRG: Ac-(RADA)4GPRGDSGYRGDS-NH_2_ KLT: Ac-(RADA)4G4KLTWQELYQLKYKGI-NH_2_	Scaffold for delivery of RGD and VEGF to promote cell adhesion, angiogenesis, and mineralization	[34,35]
RADA16-I combined with RGD motif; with two functional motifs IKVAV and RGD	Neuron and ligament regeneration	[36,37,38,39]
RADA16I: RADARADARADARADA RADA16 II: RARADADARARADADA	Controlled drug release of lypophilic drugs; hepatocyte regeneration; neuron regeneration; osteogenesis; hemostasis application	[35,40,41,42,43,44]
V3A3E3: VVVAAAEEE	Stem cell culture and differentiation	[45,46]
FEFEFKFK and nanotube	Increase fibroblast attachment, spreading, proliferation, and movement for 3D tumor and cartilage tissue engineering	[47,48]
Q11: QQKFQFQFEQQ	Vaccine platform to elicit potent HPV antigen	[49]
Ac-GRGDPS-GG-FKFEFKFE-CONH_2_	Control of matrix adhesiveness and stiffness of extracellular matrix	[50]
SPG-178 (Self-assembling Peptide Gel, amino acid sequence): Ac-RLDLRLALRLDLR-NH_2_	Scaffold for regeneration bone/nervous matrix	[51,52]
KLD12: KFDLKKDLKLDL	Tissue engineering combined with antimicrobial properties; treatment of articular cartilage	[53,54]
fFLDV	Extracellular matrix mimicry of fibronectin to lead to high cell viability, adhesion, and spreading	[55]
fF	Support for fibroblast cell proliferation and viability	[56]
(LDLK)_3_	Branched peptide with enhanced mechanical properties for tissue engineering	[57]
FEFQFK	Drug release of fluorescein sodium and ciprofloxacin hydrochloride	[58]
EAK16II: AEAEAKAKAEAEAKAK	Delivery system for peptide based vaccines for infectious diseases	[59]
BocFFF	Cancer drug release	[60]
LFF	Drug release of ciprofloxacin for antimicrobial applications	[61]

## Data Availability

Not applicable.

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
