# Peer review of "Peptide–Protein Interactions: From Drug Design to Supramolecular Biomaterials"

_molecules, 2021, doi:10.3390/molecules26051219_

Round 1

Reviewer 1 Report

The review focuses a hot topic, exploring in depth physicochemical properties of peptide-protein interactions and potential applications in the medical field. The references are up-to-date and covers adequately the subject.  Except for figure 4, which is not necessary, the other illustrations are OK. The text needs minor revision of English language and correction of some typos.

All references in the range of 21 to 58 are not cited in text.

Author Response

R: The review focuses a hot topic, exploring in depth physicochemical properties of peptide-protein interactions and potential applications in the medical field. The references are up-to-date and covers adequately the subject.  Except for figure 4, which is not necessary, the other illustrations are OK. The text needs minor revision of English language and correction of some typos.

All references in the range of 21 to 58 are not cited in text.

A: We thank the reviewer for the supportive comments and constructive feedback. We agree that figure 4 is not necessary, but in our opinion, it gives a clear picture to non-familiar readers of the content of section 3, thus we’d like to keep it. The topic of drug delivery is indeed very vast, and here we focus on the aspects depicted in Figure 4. All references in the range of 21-58, now 25-61, are cited in the Table 1. They now have been added also in the main text.

Reviewer 2 Report

The manuscript provides a good review of the physiochemical mechanism of self-recognized and self-assemble in different supramolecular structures. It especially addresses interactions via peptide blocks but does not exhaust the subject. The examples presented guarantee publication, however, there is no mention of the delivery of drugs of peptide nature, for example, which do not depend on nanoemulsions. Likewise, it addresses the adjuvant effect of some structures superficially, not going deeper into the immune level. The manuscript needs an English review and extensive typing corrections. Line 71 requires the definition and meaning of SAPs.

Author Response

R: The manuscript provides a good review of the physiochemical mechanism of self-recognized and self-assemble in different supramolecular structures. It especially addresses interactions via peptide blocks but does not exhaust the subject. The examples presented guarantee publication, however, there is no mention of the delivery of drugs of peptide nature, for example, which do not depend on nanoemulsions. Likewise, it addresses the adjuvant effect of some structures superficially, not going deeper into the immune level. The manuscript needs an English review and extensive typing corrections. Line 71 requires the definition and meaning of SAPs.

A1: We thank the Reviewer for the constructive feedback. The topic is indeed very vast, and many comprehensive reviews already exist, therefore our attention was to focus on specific aspects that link the design of bioactive motifs with self-assembling motifs, based on weak protein-peptide or peptide-peptide interactions. We have now added reference to a very recent review that covers the drug-delivery of proteins (Ref. 64, 65) and peptide-based therapeutics (Ref. 66,67).

A2: The manuscript needs an English review and extensive typing corrections.

A3: The definition SAP (plural SPAs) is now in line 64.

Reviewer 3 Report

Overall, this covers and interesting topic with a unique stance on a burgeoning technology.  However, the paper lacks a clear direction and there are many places where removing extraneous (and seemingly "out of nowhere" sections and instead adding more to the highly impactful sections would benefit the authors' point of view.

1) Introduction.  Through line 77, prior to the first table, the information was a lot of listing applications, not necessarily interrelated, and not expanding on examples or how they fit into the context of the paper.  In Table 1 it was unclear if the column labeled "Peptide" were representing sequences and the application column was way too vague in many cases (just simply stating "drug delivery.")  Lines 90-92 have little if anything to do with supramolecular assembly and do not appear to play a role in the continuity of this section.  At the end of the introduction the authors need some kind of main focus for the paper to come through.  There are so many myriad things explained without any real point tying it together.  There needs to be a main theme or focus, and the way this section ends with beta sheets and tissue regeneration appears to make THAT the focus, which does NOT come through when actually reading the rest of the paper.

2)  Section 2 immediately there are 5 listed strategies for peptide drug design (a-e) and yet this has no clear tie to self-assembly, which is the overarching theme of the paper.  I would remove these or least try to connect them to assembly and secondary structure.  Figure 1 the amphipathic helix (spelled incorrectly as amphipatic) does not show that it is amphipathic, it could represent any coiled coil.  A more discrete molecular look may help.  The polyproline II part of the figure also is very confusing and I was unsure what the authors were representing. In lines 130-132 GPCRs are mentioned however their description as supramolecular structures (the number of helices, how they orient in the cell membrane etc) is left out and should be included.  In line 141 there's a peptide with Aib residues included, which implies it isn't naturally found, however no context is supplied as to where this peptide came from and if it is synthetic, why it is being mentioned etc.  The entire section from lines 143-155 does not seem to connect to supramolecular structures (again, the theme of the review) at all and should either be removed or completely retooled to include context as to why this is here.  Lines 175-~200 are really excellent! Perhaps more is needed here and less of the areas that don't pertain.  Again line 208 an Aib containing peptide is included but again it's unclear as to where it came from and its context.  Lines 243-~251 again, don't really seem to relate to beta-sheets or beta-hairpins directly and it is unclear why this section is included.  Please find a way to better explain how it connects, or remove it.  What does it have to do with assembly?  Lines 275-276 need clarity - what is bioactive about that motif and where does it come from?  The description of IAPP and the figure seem a bit out of context, as well - they should perhaps be included in the prior section on  beta sheets because they're mostly about inhibiting self assembly.  Line 405 would be aided by a figure to show what an oligoproline-perylene monoimide conjugate looks like.   The whole section 2.4 was excellent and should actually have MORE emphasis and more included, with less emphasis on the other sections, particularly those that didn't seem to fit.

4) Conclusions (section 3 was great as is!) - the only problem with the conclusions is that there was no unifying takeaway from the review, no one big picture statement.  

Author Response

Overall, this covers and interesting topic with a unique stance on a burgeoning technology.  However, the paper lacks a clear direction and there are many places where removing extraneous (and seemingly "out of nowhere" sections and instead adding more to the highly impactful sections would benefit the authors' point of view.

R: 1) Introduction.  Through line 77, prior to the first table, the information was a lot of listing applications, not necessarily interrelated, and not expanding on examples or how they fit into the context of the paper.  In Table 1 it was unclear if the column labeled "Peptide" were representing sequences and the application column was way too vague in many cases (just simply stating "drug delivery.")   Lines 90-92 have little if anything to do with supramolecular assembly and do not appear to play a role in the continuity of this section.  At the end of the introduction the authors need some kind of main focus for the paper to come through.  There are so many myriad things explained without any real point tying it together.  There needs to be a main theme or focus, and the way this section ends with beta sheets and tissue regeneration appears to make THAT the focus, which does NOT come through when actually reading the rest of the paper.

A: We thank the Reviewer for the constructive feedback. We appreciate the help given to us in order to very much improve the manuscript. The Table 1, the column labeled “Peptide” is now changed in “Sequences”. In the column “application”, a more detailed description has been introduced. We re-wrote parts of Introduction. The lines 92-101 have been now moved in section 2, in order to better stress the focus of the manuscript.

2)  Section  2. immediately there are 5 listed strategies for peptide drug design (a-e) and yet this has no clear tie to self-assembly, which is the overarching theme of the paper.  I would remove these or least try to connect them to assembly and secondary structure.  Figure 1 the amphipathic helix (spelled incorrectly as amphipatic) does not show that it is amphipathic, it could represent any coiled coil.  A more discrete molecular look may help.  The polyproline II part of the figure also is very confusing and I was unsure what the authors were representing. In lines 130-132 GPCRs are mentioned however their description as supramolecular structures (the number of helices, how they orient in the cell membrane etc) is left out and should be included.  In line 141 there's a peptide with Aib residues included, which implies it isn't naturally found, however no context is supplied as to where this peptide came from and if it is synthetic, why it is being mentioned etc.  The entire section from lines 143-155 does not seem to connect to supramolecular structures (again, the theme of the review) at all and should either be removed or completely retooled to include context as to why this is here.  Lines 175-~200 are really excellent! Perhaps more is needed here and less of the areas that don't pertain.  Again line 208 an Aib containing peptide is included but again it's unclear as to where it came from and its context.  Lines 243-~251 again, don't really seem to relate to beta-sheets or beta-hairpins directly and it is unclear why this section is included.  Please find a way to better explain how it connects, or remove it.  What does it have to do with assembly?  Lines 275-276 need clarity - what is bioactive about that motif and where does it come from?  The description of IAPP and the figure seem a bit out of context, as well - they should perhaps be included in the prior section on beta sheets because they're mostly about inhibiting self-assembly.  Line 405 would be aided by a figure to show what an oligoproline-perylene monoimide conjugate looks like.    The whole section 2.4 was excellent and should actually have MORE emphasis and more included, with less emphasis on the other sections, particularly those that didn't seem to fit.

A: We thank the Reviewer for the accurate reading and we very much appreciated his/her suggestions. We removed the lines 104-115 (the list of strategies for peptide drug design (a-e)), and we introduced the description of weak forces at the basis (origin) of supramolecular architectures, previously described in lines 93-101.

Then, we modified Figure 1, as suggested by the Reviewer.

We rewrote the section on GPCR, in particular we described the supramolecular structure of the receptor and compacted the lines 141-155, removing Figure 2. Moreover, we moved lines 208-217 in order to make the text more homogeneous. We cited ref 114 in order to explain better the peptidic nature in line 141 (now in line 163)

As suggested by the Reviewer, we removed lines 243-251.

As suggested by the Reviewer, we moved the description of IAPP from section 2.3 to section 2.2.

We better described the topic reported in lines 275-276 (now lines 359-371)

As suggested by the  Reviewer , in section 2.4, we have now introduced a figure to show what an oligoproline-perylene monoimide conjugate looks like.

3) (section 3 was great as is!)

A: we thank the Reviewer

4) Conclusions - the only problem with the conclusions is that there was no unifying takeaway from the review, no one big picture statement. 

A: We agree, and have fully appreciated the observation of the Reviewer. The entire section now entitled “Conclusions and Perspectives” has been rewritten.
